

# 1  Objective Evaluation of Earth System Models: PCMDI Metrics Package

# 2  (PMP) version 3

Jiwoo Lee[1], Peter J. Gleckler[1], Min-Seop Ahn[2,3], Ana Ordonez[1], Paul A. Ullrich[1,4], Kenneth R.
Sperber[1,a], Karl E. Taylor[1], Yann Y. Planton[5,6], Eric Guilyardi[7,8], Paul Durack[1], Celine Bonfils[1],
Mark D. Zelinka[1], Li-Wei Chao[1], Bo Dong[1], Charles Doutriaux[1], Chengzhu Zhang[1], Tom Vo[1],
Jason Boutte[1], Michael F. Wehner[9], Angeline G. Pendergrass[10,11], Daehyun Kim[12], Zeyu Xue[13],
Andrew T. Wittenberg[14], and John Krasting[14]
[1] Lawrence Livermore National Laboratory, Livermore, California, USA
[2] NASA Goddard Space Flight Center, Greenbelt, MD, USA
[3] ESSIC, University of Maryland, College Park, MD, USA
[4] University of California, Davis, Davis, California, USA
[5] NOAA Pacific Marine Environmental Laboratory, Seattle, Washington, USA
[6] Monash University, Clayton, Australia
[7] LOCEAN-IPSL, CNRS-IRD-MNHN-Sorbonne Université, Paris, France
[8] National Centre for Atmospheric Science-Climate, University of Reading, Reading, UK
[9] Lawrence Berkeley National Laboratory, Berkeley, California, USA
[10] Department of Earth and Atmospheric Science, Cornell University, Ithaca, New York, USA
[11] National Center for Atmospheric Research, Boulder, Colorado, USA
[12] School of Earth and Environmental Sciences, Seoul National University, Seoul, South Korea
[13] Pacific Northwest National Laboratory, Richland, WA, USA
[14] NOAA Geophysical Fluid Dynamics Laboratory, Princeton, NJ, USA
[a] Retired
*Corresponding to:* Jiwoo Lee (lee1043@llnl.gov)
7000 East Ave, Livermore, California 94550, USA



**Abstract**
Systematic, routine, and comprehensive evaluation of Earth System Models (ESMs) facilitates benchmarking
improvement across model generations and identifying the strengths and weaknesses of different model
configurations. By gauging the consistency between models and observations, this endeavor is becoming increasingly
necessary to objectively synthesize thousands of simulations contributed to the Coupled Model Intercomparison
Project (CMIP) to date. The PCMDI Metrics Package (PMP) is an open-source Python software package that provides
"quick-look" objective comparisons of ESMs with one another and with observations. The comparisons include
metrics of large- to global-scale climatologies, tropical inter-annual and intra-seasonal variability modes such as El
Niño-Southern Oscillation (ENSO) and Madden-Julian Oscillation (MJO), extratropical modes of variability, regional
monsoons, cloud radiative feedbacks, and high-frequency characteristics of simulated precipitation, including
extremes. The PMP results are produced in the context of all model simulations contributed to CMIP6 and earlier
CMIP phases. An important priority of the PMP is to document evaluation statistics for all Historical and AMIP
simulations submitted to recent phases of CMIP, providing version-controlled information for all data sets and
software packages being used. Among other purposes, this also enables modeling groups to assess performance
changes during the ESM development cycle in the context of the error distribution of the multi-model ensemble. In
this paper, we present an overview of the PMP including its history to date, capabilities, recent updates, and future
direction.



## 1 Introduction

Earth System Models (ESMs) are key tools for projecting climate change and conducting research to enhance our understanding of the Earth system. Enhancing the reliability of models is therefore important, yet evaluating ESMs is a complex endeavor, given the vast range of climate characteristics across space and time scales. A necessary step to evaluate the performance of ESMs is quantifying their consistency with available observations.

The Program for Climate Model Diagnosis and Intercomparison (PCMDI) has worked closely with the World Climate Research Programme's (WCRP) Working Group on Coupled Models (WGCM) and Working Group on Numerical Experimentation (WGNE) to design and support Model Intercomparison Projects (MIPs) (Potter et al., 2011). This effort began with the Atmospheric Model Intercomparison Project (AMIP; Gates, 1992; Gates et al., 1999), and has continued through multiple phases of the Coupled Model Intercomparison Project (CMIP; Meehl et al., 1997, 2000, 2007; Covey et al., 2003; Taylor et al., 2012). The most recent phase of CMIP (CMIP6; Eyring et al., 2016) provides a set of well-defined experiments that most climate modeling centers perform, and subsequently makes results available for a large and diverse community to analyze.

Climate model performance metrics have been widely used to objectively and quantitatively gauge the agreement between observations and simulations to summarize model behavior in a wide range of model evaluations. Simple examples include either the model bias or the pattern similarity (correlation) between an observed and simulated field (e.g., Taylor, 2001). With the rapid growth in the number, scale, and complexity of simulations, the metrics have been used more routinely as exemplified by the Intergovernmental Panel on Climate Change (IPCC) Assessment Reports (e.g., Gates et al., 1995; McAvaney et al., 2001; Randall et al., 2007; Flato et al., 2014; Eyring et al., 2021). A few studies have been exclusively devoted to objective model performance assessment using summary statistics. Lambert and Boer (2001) evaluated the first set of CMIP models from CMIP1 using statistics for the large-scale mean climate. Gleckler et al. (2008) identified a variety of factors relevant to model metrics and demonstrated techniques to quantify the relative strengths and weaknesses of the simulated mean climate. Reichler and Kim (2008) attempted to gauge model improvements across the early phases of CMIP. The scope of objective model evaluation has greatly broadened beyond the mean state in recent years (e.g., Gleckler et al., 2016; Eyring et al., 2019), including attempts to establish performance metrics for a wide range of climate variability (e.g., Kim et al., 2009; Sperber et al., 2013; Ahn et al., 2017; Fasullo et al., 2020; Lee et al., 2021b; Planton et al., 2021) and extremes (e.g., Sillmann et al., 2013; Srivastava et al., 2020; Wehner et al., 2020, 2021). Guilyardi et al. (2009) and Reed et al. (2022) emphasized that metrics should be concise, interpretable, informative, and intuitive.

Considering the exponential growth of data size and diversity of ESM simulations, there has been a pressing need for the research community to become more efficient and systematic in evaluating ESMs and documenting their performances. To respond to the need, PCMDI has developed the PCMDI Metrics Package (PMP), to quantitatively synthesize results from the archive of CMIP simulations via performance metrics that help characterize the overall agreement between models and observations (Gleckler et al., 2016). In this paper, we describe the latest update of the PMP and its focus on providing a diverse suite of summary statistics that can be used to construct "quick-look" summaries of ESM performance from simulations made publicly available to the research community, notably CMIP. For our purposes, "performance metrics" are typically (but not exclusively) well-established statistical measures that



quantify the consistency between observed and simulated characteristics. One goal of the PMP is to further diversify
the suite of high-level performance tests that help characterize the simulated climate. The results provided by the PMP
are frequently used to address two overarching and recurring questions: 1) What are the relative strengths and
weaknesses between different models? and 2) How are models improving with further development? Addressing the
second question is often referred to as "benchmarking" and this motivates an important emphasis of the effort
described in this paper—striving to advance the documentation of all data and results of the PMP in an open and
ultimately reproducible manner.
The rest of the paper is organized as follows. In section 2, we provide a technical description of the PMP and
its accompanying reference datasets. In section 3, we describe various sets of simulation metrics that capture an
increasingly comprehensive range of physical processes and time scales ranging from hours to centurial. In section 4,
we introduce the usage of PMP for model benchmarking. In section 5, we discuss the remaining challenges, and we
conclude in section 6 with a summary and future direction.

## 2 Software package and data description

The PMP is a Python-based open-source software framework (https://github.com/PCMDI/pcmdi_metrics) designed
to objectively gauge the consistency between ESMs and available observations via well-established statistics. The
PMP has been mainly used for the evaluation of CMIP-class models. A subset of CMIP experiments are particularly
well suited to comparing models with observations. The experiments of particular interest include those involving
prescribed sea surface temperature (SST) in accordance with the AMIP protocol, as well as coupled model simulations
labeled as "Historical" that are driven by varying natural and anthropogenic forcings. Some of the metrics applicable
to these experiments may also be relevant to others (e.g., multi-century coupled control runs called "PiControl" and
idealized "4xCO2" simulations that are designed for estimating climate sensitivity).
The PMP has been applied to multiple generations of CMIP in a quasi-operational fashion as new simulations
are made available, new analysis methods are incorporated, or new observational data become accessible. Shortly
after simulations from the most recent phase of the CMIP (i.e., CMIP6) became accessible, PMP quick-look
summaries were provided on the PCMDI's website (https://pcmdi.llnl.gov/metrics/), offering a resource to scientists
involved in CMIP or others interested in the evaluation of ESMs. To facilitate this, in PCMDI the PMP is technically
linked to the Earth System Grid Federation (ESGF) that is a primary CMIP data delivery infrastructure (Williams et
al., 2016).
The PMP is designed to readily work with model output that has been processed using the Climate Model
Output Rewriter (CMOR; https://cmor.llnl.gov/), which is a software library developed to prepare model output as
CF-compliant (Hassell et al., 2017; Eaton et al., 2022, http://cfconventions.org/) netCDF files. The CMOR is used by
most modeling groups contributing to CMIP, ensuring all model output adheres to the CMIP data structures that
themselves are based on the CF conventions. It is possible to use the PMP on model output that has not been prepared
by CMOR, but this usually requires additional work, e.g., mapping the data to meet the community standards.
For reference datasets, the PMP uses observational products processed to be compliant with the Observations
for Model Intercomparison Projects (obs4MIPs; https://pcmdi.github.io/obs4MIPs/). The obs4MIPs effort was



initiated circa 2010 (Gleckler et al., 2011) to advance the use of the observations in model evaluation and research.
Substantial progress has been made in establishing obs4MIPs data standards that technically align with CMIP model
output (e.g., Teixeira et al., 2014; Ferraro et al., 2015), with the data products published on the ESGF (Waliser et al.,
2020). Obs4MIPs-compliant data were prepared with CMOR, and the data directly available via obs4MIPs are used
as PMP reference datasets.

The PMP leverages other Python-based open-source libraries. A primary fundamental tool used in the latest
PMP version is the Python package, Xarray Climate Data Analysis Tools (xCDAT; Vo et al., 2023;
https://xcdat.readthedocs.io). The xCDAT is developed to provide a more efficient, robust, and streamlined user
experience in climate data analysis when using xarray (https://docs.xarray.dev/). Portions of the PMP rely on the
precursor of the xCDAT, a Python library called Community Data Analysis Tools (CDAT, Williams et al., 2009;
Williams, 2014; Doutriaux et al., 2019), which has been fundamental since the early development stages of the PMP.
The xarray software provides much of the functionality of CDAT (e.g., I/O, indexing, and subsetting). However, it
lacks some key climate domain features that have been frequently used by scientists and exploited by the PMP (e.g.,
regridding, utilization of spatial/temporal bounds for computational operations) which motivated the development of
the xCDAT. Completing the transition from CDAT to xCDAT is a technical priority for the next version of PMP.

The primary delivery output of the PMP is the summary statistics. We strive to make the baseline results (raw
statistics) publicly available and well-documented, and continue to make advances with this priority. For our purposes,
we are referring to model performance "summary statistics" and "metrics" interchangeably, although in some
situations we consider there to be an important distinction. For us, a genuine performance metric constitutes a well-
defined and established statistic that has been used in a very specific way (e.g., a particular variable, analysis, and
domain) for long-term benchmarking (see Section 4). The distinction between summary statistics and metrics is
application-dependent and evolving as the community advances efforts to establish quasi-operational capabilities to
gauge ESM performance. Some visualization capabilities described in Section 3 are made available through the PMP.
Users can also further explore the model data comparisons using their preferred visualization methods or incorporate
the results into their own studies from the summary statistics from the PMP. Noting the above, the scope of the PMP
is fairly targeted. It is not intended to be "all-purpose", e.g. by incorporating the vast range of diagnostics used in
model evaluation.

To help advance open and reproducible science, the PMP has been maintained with an open-source policy
with accompanying metadata for data reproducibility and reusability. The PMP code is distributed and released with
version control. Online documentation (http://pcmdi.github.io/pcmdi_metrics/), including user demo Jupyter
Notebooks, and a database of pre-calculated PMP statistics for all AMIP and Historical simulations in the CMIP
archive are also available online. The archive of these statistics stored as JSON files (Crockford, 2006; Crockford and
Morningstar, 2017) includes versioning details for all codes, and dependencies and data that were used for the
calculations. These files provide the baseline results of the PMP (See the Code and Data Availability section for
details). Advancements in model evaluation along with the number of models and complexity of simulations motivate
more systematic documentation of performance summaries. With PMP workflow provenance information being



recorded and the model and observational data standards maintained by PCMDI and colleagues, PMP strives to make
all its results reproducible.

**3 Current PMP capabilities**

The PMP builds upon model performance tests that have resulted from research at PCMDI and via close
collaborations. Contributors have helped expand the PMP beyond its traditional large-scale performance summaries
of the mean climate (Gleckler et al., 2008). Various evaluation metrics have been implemented to the PMP for climate
variability such as El Niño-Southern Oscillation (ENSO) (Planton et al., 2021; Lee et al., 2021a), extratropical modes
of variability (Lee et al., 2019, 2021b), intra-seasonal oscillation (Ahn et al., 2017), monsoons (Sperber and
Annamalai, 2014), cloud feedback (Zelinka et al., 2022), and the characteristics of simulated precipitation
(Pendergrass et al., 2020; Ahn et al., 2022, 2023) and extremes (Wehner et al., 2020, 2021). This section will provide
an overview of each category of the current PMP evaluation metrics with their usage demonstrations.

*3.1 Climatology*

Mean state metrics quantify how well models simulate observed climatological fields at a large scale, gauged by a
suite of well-established statistics that have been used in climate research for decades. The focus is on the coupled
"Historical" and atmospheric-only AMIP (Gates et al., 1999) simulations which are well-suited for comparison with
observations. The PMP extracts seasonally and annually averaged fields of multiple variables from large-scale
observationally based datasets and results from model simulations. Different obs4MIPs-compliant reference datasets
are used depending on the variable examined. When multiple reference datasets are available, one of them is
considered as a "default" while others are identified as "alternatives". The default datasets are typically state-of-the-
art products, but in general, we lack definitive measures as to which is the most accurate, so the PMP metrics are
routinely calculated with multiple products so that it can be determined what difference the selection of alternative
observations makes to judgment made about model fidelity. The suite of mean climate metrics (all area weighted)
includes spatial and spatiotemporal root-mean-square error (RMSE), centered spatial RMSE, spatial-mean bias, spatial
standard deviation, spatial pattern correlation, and spatial and spatiotemporal mean absolute error (MAE) of the annual
or seasonal climatological time-mean (Gleckler et al., 2008). Often, a space-time statistic is used that gauges both the
consistency of the observed and simulated climatological pattern as well as its seasonal evolution (see Eq. 1 from
Gleckler et al., 2008). By default, results are available for selected large-scale domains, including: "Global", "Northern
Hemisphere (NH) Extratropics" (30ºN-90ºN), "Tropics" (30ºS-30ºN), and "Southern Hemisphere (SH) Extratropics"
(30ºS-90ºS). For each domain, results can also be computed for the land and ocean, land only, or ocean only. These
commonly used domains highlight the application of the PMP mean climate statistics at large to global scales, but we
note that PMP allows users to define their own domains of interest, including at regional scales.
Although the primary deliverable of the PMP is the metrics, these PMP results can be visualized in various
ways. For individual fields, we often first plot Taylor Diagrams, a polar plot leveraging the relationship between the
centered RMS, the pattern correlation, and the observed and simulated standard deviation (Taylor, 2001). The Taylor
Diagram has become a standard plot in the model evaluation workflow across modeling centers and research



communities (see Section 5). To interpret results across CMIP models for many variables, we routinely construct
normalized Portrait Plots or Gleckler Plots (Gleckler et al., 2008) that provide a quick-look examination of the
strengths and weaknesses of different models. For example, in Figure 1, the PMP results display quantitative
information of simulated seasonal climatologies of various meteorological model variables via a normalized global
spatial RMSE (Gleckler et al., 2008). Variants of this plot have been widely used for presenting model evaluation
results, for example, in the Intergovernmental Panel on Climate Change (IPCC) Fifth (Flato et al., 2014, Figures 9.7,
9.12, and 9.37) and Sixth Assessment Reports (Eyring et al., 2021, Chapter 3, Figure 3.42). Because the error
distribution across models is variable dependent, the statistics are often normalized to help reveal differences, in this
case via the median RMSE across all models (see Gleckler et al. 2008 for more details). This normalization enables a
common color scale to be used for all statistics on the Portrait Plot, highlighting the relative strengths and weaknesses
of different models. In this example (Fig. 1), an error of -0.5 indicates that a model's error is 50% smaller than the
typical (median) error across all models, whereas an error of 0.5 is 50% larger than the typical error in the multi-model
ensemble. In many cases, the horizontal bands in the Gleckler plots show that simulations from a given modeling
center have similar error structures relative to the multi-model ensemble.
The Parallel Coordinate Plot (Inselberg, 1997, 2008, 2016; Johansson and Forsell, 2016) that retains the
absolute value of the error statistics is used to complement the Portrait plot. Some previous studies have utilized
Parallel Coordinate Plots for analyzing climate model simulations (e.g., Steed et al., 2012; Wong et al., 2014; Wang
et al., 2017), but to date, only a few studies have applied it to collective multi-ESM evaluations (e.g., see Fig. 7 of
Boucher et al., 2020). In the PMP, we generally construct Parallel Coordinate Plots using the same data as in a portrait
plot. However, a fundamental difference is that metrics values can be more easily scaled to highlight absolute values
rather than the normalized relative results of the portrait plot. In this way, the Portrait and Parallel Coordinate plots
complement one another, and in some applications, it can be instructive to display both. Figure 2 shows the
spatiotemporal RMSE, defined as the temporal average of spatial RMSE calculated in each month of the annual cycle,
of CMIP5 and CMIP6 models in the format of Parallel Coordinate Plot. Each vertical axis represents a different scalar
measure gauging a distinct aspect of model fidelity. While polylines are frequently used to connect data points from
the same source (i.e., metric values from the same model, in our case) in Parallel Coordinate Plots, we display results
from each model using an identification symbol to reduce visual clutter on the plot and help identify outlier models.
In the example of Fig. 2, each vertical axis is aligned with the median value midway through its max/min range scale.
Thus, for each axis, the models in the lower half of the plot perform better than the CMIP5-CMIP6 multi-model
median, while in the upper half, the opposite is true. For each vertical axis that is for a different model variable, we
have added violin plots (Hintze and Nelson, 1998) to show probability density functions representing the distributions
of model performance obtained from CMIP5 (shaded in blue, left side of the axis) and CMIP6 (shaded in orange, right
side of the axis). Medians of each CMIP5 and CMIP6 group are highlighted using polylines, which indicates that the
RMSE is reduced in CMIP6 relative to CMIP5 in general for the majority of the subset of model variables.



*3.2 El Niño-Southern Oscillation*


The El Niño-Southern Oscillation (ENSO) is Earth's dominant interannual mode of climate variability, which impacts
global climate via both regional oceanic effects and far-reaching atmospheric teleconnections (McPhaden et al., 2006,
2020). In response to increasing interest in a community approach to ENSO evaluation in models (Bellenger et al.,
2014), the International Climate and Ocean Variability, Predictability and Change (CLIVAR) Research Focus on
ENSO in a Changing Climate, together with the CLIVAR Pacific Region Panel, developed the CLIVAR ENSO
Metrics Package (Planton et al., 2021) which is now utilized within the PMP. The ENSO metrics are divided into three
Metrics Collections: *Performance* (i.e., background climatology and basic ENSO characteristics), *Teleconnections*
(ENSO's worldwide teleconnections), and *Processes* (ENSO's internal processes and feedback). Planton et al. (2021)
found that CMIP6 models generally outperform CMIP5 models in several ENSO metrics in particular for those related
to tropical Pacific seasonal cycles and ENSO teleconnections. This effort is discussed in more detail in Planton et al.
(2021), and detailed descriptions of each metric in the package are available in the ENSO Package online open-source
code repository on its GitHub Wiki pages (see https://github.com/CLIVAR-PRP/ENSO_metrics/wiki).
Figure 3 demonstrates the application of the ENSO metrics to CMIP6, showing the magnitudes of inter-
model and inter-ensemble spreads, along with observational uncertainty varying across metrics. For a majority of the
ENSO Performance metrics model error and inter-model spread are substantially larger than observational uncertainty
(Figs. 3a-n). This highlights the systematic biases like the double ITCZ (Fig. 3a) that are persisting through CMIP
phases (Tian and Dong, 2020). Similarly, ENSO Processes metrics (Figs. 3t-w) indicate large errors in the feedback
loops generating SST anomalies, indicating a different balance of processes in the model and in the reference and
possibly compensating errors (Bayr et al., 2019, Guilyardi et al. 2020). In contrast, for ENSO Teleconnection metrics,
the observational uncertainty is substantially larger, thus challenging validation of model error (Figs. 3o-r). For some
metrics, such as the ENSO duration (Fig. 3f), the ENSO Asymmetry metric (Fig. 3i), and the Ocean driven SST metric
(Fig. 3s), there are larger inter-ensemble spreads than the inter-model spreads. From such results, Lee et al. (2021a)
examined the inter-model and inter-member spread of these metrics from the large ensembles available from CMIP6
and the US CLIVAR Large Ensemble Working Group. They argued that to robustly characterize baseline ENSO
characteristics and physical processes, larger ensemble sizes are needed, compared to existing state-of-the-art
ensemble projects.

*3.3 Extratropical Modes of Variability*


The PMP includes objective measures of the pattern and amplitude of extratropical modes of variability from
PCMDI's research, which has expanded beyond its traditional large-scale performance summaries to include
interannual variability, considering increasing interest in setting an objective approach for the collective evaluation of
multiple modes. Extratropical modes of variability (ETMoV) metrics in the PMP were developed by Lee et al. (2019a)
that stem from earlier works (e.g., Stoner et al., 2009; Phillips et al., 2014). Lee et al. (2019a) illustrated a challenge
when evaluating modes of variability using the traditional empirical orthogonal functions (EOF). In particular, when
a higher-order EOF of a model more closely corresponds to a lower-order observationally based EOF (or vice versa),
it can significantly affect conclusions drawn about model performance. To circumvent this issue in evaluating the



interannual variability modes, Lee et al. (2019a) used the Common Basis Function (CBF) approach that projects the
observed EOF pattern onto model anomalies. This approach has been previously applied for the evaluation of
intraseasonal variability modes (Sperber, 2004; Sperber et al., 2005), and recently for Antarctic climate change (Jun
et al., 2020), seasonal-to-decadal predictability associated with the ENSO (Choi and Son, 2022). In the PMP, the CBF
approach is taken as a default method, and the traditional EOF approach is also enabled as an option for the ETMoV
metrics calculations.
The ETMoV metrics in the PMP measure simulated patterns and amplitudes of ETMoV, and quantify their
agreement with observations (e.g., Lee et al., 2019a, 2021b). The PMP's ETMoV metrics evaluate 5 atmospheric
modes – the Northern Annular Mode (NAM), North Atlantic Oscillation (NAO), Pacific North America pattern
(PNA), North Pacific Oscillation (NPO), and Southern Annular Mode (SAM), and 3 ocean modes diagnosed by the
variance of sea-surface temperature – Pacific Decadal Oscillation (PDO), North Pacific Gyre Oscillation (NPGO),
and Atlantic Multi-decadal Oscillation (AMO). The AMO is included for experimental purposes, considering the
significant uncertainty in detecting the AMO (Deser and Philips 2021; Zhao et al., 2022). The amplitude metric,
defined as the ratio of standard deviations of the model and observed principal components, has been used to examine
the evolution of the performance of models across different CMIP generations (Fig. 4, adapted from Lee et al., 2021b).
Green shading predominates, indicating where the simulated amplitude of variability is similar to observations. In
some cases, such as for SAM_SON, the models overestimate the observed amplitude. Other authors have used Portrait
plots to synthesize CMIP performance of simulated variability (e.g., Sillmann et al., 2013; Bellenger et al., 2014;
Cannon 2020; Kim et al., 2020; Planton et al., 2020; Zhang et al., 2021; Ahn et al., 2022, 2023).
The PMP's ETMoV metrics have been used in several model evaluation studies. For example, Orbe et al.
(2020) analyzed models from U.S. climate modeling groups including DOE, National Aeronautics and Space
Administration (NASA), National Center for Atmospheric Research (NCAR), and National Oceanic and Atmospheric
Administration (NOAA), where they found that the improvement in the ETMoV performance is highly dependent on
mode and season, when comparing across different generations of those models. Sung et al. (2021) examined the
performance of models run at the Korea Meteorological Administration (K-ACE and UKESM1) in reproducing
ETMoVs from their Historical simulations, and concluded that these models reasonably capture most ETMoVs. Lee
et al. (2021b) collectively evaluated ~130 models from CMIP3, 5, and 6 archive databases using their ~850 Historical
and ~300 AMIP simulations, where they found the spatial pattern skill improved in CMIP6 compared to CMIP5 or
CMIP3 for most modes and seasons, while the improvement in amplitude skill is not clear. Arcodia et al. (2023) used
the PMP to derive PDO and AMO to investigate their role in decadal variability of subseasonal predictability of
precipitation over the western coast of North America and concluded that no significant relationship was found.

*3.4 Intraseasonal Oscillation*
The PMP has implemented metrics for the Madden-Julian Oscillation (MJO; Madden and Julian, 1971, 1972, 1994).
The MJO is the dominant mode of tropical intraseasonal variability, characterized by a pronounced eastward
propagation of large-scale atmospheric circulation coupled with convection with a typical periodicity of 30-60 days.



304 Selected metrics from the MJO diagnostics package, developed by the CLIVAR MJO Working Group (Waliser et al.,
305 2009), have been implemented in the PMP following Ahn et al. (2017).

306   We particularly focused on a metric called East/West power Ratio (hereafter, EWR) and East power
307 normalized by Observation (hereafter, EOR). The EWR, proposed by Zhang and Hendon (1997), is defined as the
308 ratio of the total spectral power over the MJO band (eastward propagating, wavenumber 1-3 and period of 30-60 days)
309 to that of its westward propagating counterpart in the wavenumber-frequency power spectra. The EWR metric has
310 been widely used in the community, to examine the robustness of the eastward propagating feature of the MJO (e.g.,
311 Zhang and Hendon, 1997; Hendon et al., 1999; Lin et al., 2006; Kim et al., 2009; Ahn et al., 2017). The EOR is
312 formulated by normalizing a model's spectral power within the MJO band by the corresponding observed value. Ahn
313 et al. (2017) showed EWRs and EORs of the CMIP5 models. Using daily precipitation, the PMP calculates EWR and
314 EOR separately for boreal winter (November to April) and boreal summer (March to October). We apply the
315 frequency-wavenumber decomposition method to precipitation from observations (GPCP-based; 1997-2010) and the
316 CMIP5 and CMIP6 Historical simulations for 1985-2004. For disturbances with wavenumbers 1-3 and frequencies
317 corresponding to 30-60 days, it is clear in observations that the eastward propagating signal dominates over its
318 westward propagating counterpart with an EWR value of about 2.49 (Fig. 5a). Figure 5b shows the wavenumber-
319 frequency power spectrum from CMIP5 IPSL-CM5B-LR as an example, which has an EWR value that is comparable
320 to the observed value.

321   Figure 6 shows the EWR from individual models' multiple ensemble members and their average. The average
322 EWR of the CMIP6 model simulations is more realistic than that of the CMIP5 models. Interestingly, a substantial
323 spread exists across models and also among ensemble members of a single model. For example, while the average
324 EWR value for the CESM2 ensemble is 2.47 (close to 2.49 from GPCP observations), the EWR values of the
325 individual ensemble members range from 1.87 to 3.23. Kang et al. (2020) suggested that the ensemble spread in the
326 propagation characteristics of the MJO can be attributed to the differences in the moisture mean state, especially its
327 meridional moisture gradient. A cautionary note should be given to the fact that the MJO frequency and wavenumber
328 windows are chosen to capture the spectral peak in observations. Thus, while the EWR provides an initial evaluation
329 of the propagation characteristics of the observed and simulated MJO, it is instructive to look at the frequency-
330 wavenumber spectra, as in some cases the dominant periodicity and wavenumber in a model may be different than in
331 observations. It is worthwhile to note that the PMP can be used to obtain EWR and EOR of other daily variables for
332 MJO analysis, such as outgoing longwave radiation (OLR) or zonal wind at 850 hPa (U-850) or 250 hPa (U-250), as
333 shown in Ahn et al. (2017).

### 3.5 Monsoons

336 Based on the work of Sperber and Annamalai (2014), skill metrics in the PMP quantify how well models represent
337 the onset, decay, and duration of regional monsoons. From observations and Historical simulations, the climatological
338 pentads of precipitation are area-averaged for six monsoon-related domains: All-India Rainfall, Sahel, Gulf of Guinea,
339 North American Monsoon, South American Monsoon, and Northern Australia, as seen in Fig. 7. For the domains in
340 the Northern Hemisphere, the 73 climatological pentads run from January to December, while for the domains in the





Southern Hemisphere, the pentads run from July to June. For each domain, the precipitation is accumulated at each
subsequent pentad and then divided by the total precipitation to give the fractional accumulation of precipitation as a
function of pentad. Thus, the annual cycle behavior is evaluated irrespective of whether a model has a dry or wet bias.
Except for GoG, the onset and decay of monsoon occur for a fractional accumulation of 0.2 and 0.8, respectively.
Between these fractional accumulations, the accumulation of precipitation is nearly linear as the monsoon season
progresses. Comparison of the simulated and observed onset, duration, and decay are presented in terms of the
difference in the pentad index obtained from the model and observations (i.e., model minus observations). Therefore,
negative values indicate that the onset or decay in the model occurs earlier than in observations, while positive values
indicate the opposite. For duration, negative values indicate that for the model it takes fewer pentads to progress from
onset to decay compared to observations (i.e., the simulated monsoon period is too short), while positive values
indicate the opposite.
For CMIP5, we find systematic errors in the phase of the annual cycle of rainfall. The models are delayed in
the onset of summer rainfall over India, the Gulf of Guinea, and the South American Monsoon, with early onset
prevalent for the Sahel and the North American Monsoon. The lack of consistency in the phase error across all domains
suggests that a ''global'' approach to the study of monsoons may not be sufficient to rectify the regional differences.
Rather, regional process studies are necessary for diagnosing the underlying causes of the regionally specific
systematic model biases over the different monsoon domains. Assessment of the monsoon fidelity in CMIP6 models
using the PMP is in progress.

*3.6 Cloud feedback and mean-state*
Uncertainties in cloud feedback are the primary driver of model-to-model differences in climate sensitivity – the global
temperature response to a doubling of atmospheric $CO_2$. Recently, an expert synthesis of several lines of evidence
spanning theory, high-resolution models, and observations was conducted to establish quantitative benchmark values
(and uncertainty ranges) for several key cloud feedback mechanisms. The assessed feedbacks are those due to changes
in high-cloud altitude, tropical marine low-cloud amount, tropical anvil cloud area, land cloud amount, middle latitude
marine low-cloud amount, and high latitude low-cloud optical depth. The sum of these six components yields the total
assessed cloud feedback, which is part of the overall radiative feedback that fed into the Bayesian calculation of
climate sensitivity in Sherwood et al. (2020). Zelinka et al. (2022) estimated these same feedback components in
climate models and evaluated them against the expert-judgment values determined in Sherwood et al. (2020),
ultimately deriving a root mean square error metric that quantifies the overall match between each model's cloud
feedback and those determined through expert judgment.
Figure 8 shows the model-simulated values for each individual feedback computed in *amip-p4K* simulations
as part of CMIP5 and CMIP6 alongside the expert judgment values. Each model is color-coded by its equilibrium
climate sensitivity (determined using *abrupt-4CO2* simulations as described in Zelinka et al., 2020), and the values
from an illustrative model (GFDL-CM4) are highlighted. Among the key results apparent from this figure is that
models typically underestimate the strength of both positive tropical marine low-cloud feedback and the negative anvil
cloud feedback relative to the central expert assessed value. The sum of all six assessed feedback components is



positive in all but two models, with a multimodel mean value that is close to the expert-assessed value, but exhibits
substantial intermodel spread.

In addition to evaluating the ability of models to match the assessed cloud feedback components, Zelinka et
al. (2022) investigated whether models with less erroneous mean-state clouds tend to have smaller errors in their
overall cloud feedback RMSE. This involved computing the mean-state cloud property error metric developed by
Klein et al. (2013). This error metric quantifies the spatiotemporal error in climatological cloud properties for clouds
with optical depths greater than 3.6, weighted by their net TOA radiative impact. The observational baseline against
which the models are compared comes from the ISCCP HGG dataset (Young et al., 2018). Zelinka et al. (2022)
showed that models with smaller mean-state cloud errors tend to have stronger but not necessarily better (less
erroneous) cloud feedback, which suggests that improving mean-state cloud properties does not guarantee
improvement in the cloud response to warming. However, the models with the smallest errors in cloud feedback tend
to also have less erroneous mean-state cloud properties, and no models with poor mean-state cloud properties have
feedback in good agreement with expert judgment.

The PMP implementation of this code computes cloud feedback by differencing fields from *amip-p4K* and
*amip* experiments and normalizing by the corresponding global mean surface temperature change rather than from
differencing *abrupt-4xCO2* and *piControl* experiments and computing feedback via regression (as was done in Zelinka
et al., 2022). This choice is made to reduce the computational burden and also because cloud feedbacks derived from
these simpler atmosphere-only simulations have been shown to closely match those derived from fully coupled
quadrupled CO2 simulations (Qin et al., 2022). The code produces figures in which the user-specified model results
are highlighted and placed in the context of the CMIP5 and CMIP6 multi-model results (e.g., Figure 8).

### *3.7 Precipitation*

Recognizing the importance of accurately simulating precipitation in ESMs and a lack of objective and systematic
benchmarking for it, and motivated by discussions with WGNE and WGCM working groups of WCRP, the DOE has
initiated an effort to establish a pathway to help modelers gauge improvement (U.S. DOE, 2020). The 2019 DOE
workshop "Benchmarking Simulated Precipitation in Earth System Models" generated two sets of precipitation
metrics: *baseline* and *exploratory* metrics (Pendergrass et al., 2020). In the PMP, we have focused on implementing
the *baseline* metrics for benchmarking simulated precipitation. In parallel, a set of *exploratory* metrics that could be
added to metrics suites including PMP in the future was illustrated by Leung et al. (2022) to extend the evaluation
scope to include process-oriented and phenomena-based diagnostics and metrics.

The *baseline* metrics gauge the consistency between ESMs and observations, focusing on the holistic set of
observed rainfall characteristics (Fig. 9). For example, the spatial distribution of mean state precipitation and seasonal
cycle are outcomes of the PMP's Climatology metrics (described in Section 3.1), which provides collective evaluation
statistics such as RMSE, standard deviation, and pattern correlation over various domains (e.g., global, NH and SH
extratropics, and Tropics, with each domain as a whole, and over land and ocean, in separate). Evaluation of
precipitation variability across many timescales with PMP is documented in Ahn et al. (2022); we summarize some
of the findings here. The precipitation variability metric measures forced (diurnal and annual cycles) and internal



variability across timescales (subdaily, synoptic, subseasonal, seasonal, and interannual) in a framework based on
power spectra of 3-hourly total and anomaly precipitation. Overall, CMIP5 and CMIP6 models underestimate the
internal variability, which is more pronounced in the higher frequency variability, while they overestimate the forced
variability (Fig. 10). For the diurnal cycle, PMP includes metrics from Covey et al. (2016). Additionally, the intensity
and distribution of precipitation are assessed following Ahn et al. (2023). Extreme daily precipitation indices and their
20-year return values are calculated using a non-stationary Generalized Extreme Value statistical method. From the
CMIP5 and CMIP6 historical simulations we evaluate model performance of these indices and their return values in
comparison with gridded land-based daily observations. Using this approach, Wehner et al. (2020) found that at
models' standard resolutions, no meaningful differences were found between the two generations of CMIP models.
Wehner et al. (2021) extended the evaluations of simulated extreme precipitation to seasonal 3-hourly precipitation
extremes produced by available HighResMIP models and concluded that the improvement is minimal with the models'
increased spatial resolutions. They also noted that the order of operations of regridding and calculating extremes
affects the ability of models to reproduce observations. Drought metrics developed by Xue and Ullrich (2021) are not
implemented in PMP directly, but are wrapped by the Coordinated Model Evaluation Capabilities (CMEC; Ordonez
et al. 2021), which is a parallel framework for supporting community-developed evaluation packages. Together, these
metrics provide a streamlined workflow for running the entire baseline metrics via the PMP and CMEC that is ready
for use by operational centers and in the CMIP7.

### 3.8 Relating metrics to underlying diagnostics

Considering the extensive collection of information generated from the PMP, efforts have supported improved
visualizations of metrics using interactive graphic user interfaces. These capabilities can facilitate the interpretation
and synthesis of vast amounts of information associated with the diverse metrics and the underlying diagnostics from
which they were derived. Via the interactive navigation interface, we can explore the underlying diagnostics behind
the PMP's summary plots. On the PCMDI website, we provide interactive graphical interfaces to enable navigating
the supporting plots to the underlying diagnostics of each model's ensemble members and their average. For example,
on the interactive mean climate plots (https://pcmdi.llnl.gov/metrics/mean_clim/), hovering the mouse cursor over a
square or triangle in the Portrait Plot, or over the markers or lines in the Parallel Coordinate Plot, reveals the diagnostic
plot from which the metrics were generated. It allows the user to toggle between several metrics (e.g., RMSE, bias,
and correlation) and regions (e.g., global, Northern/Southern Hemisphere, and Tropics), along with relevant
provenance information. Users can click on the interactive plots to get dive-down diagnostics information for the
model of interest which provides detailed analysis to better understand how the metric was calculated. As with the
PMP's mean climate metrics output, we currently provide interactive summary graphics for ENSO
(https://pcmdi.llnl.gov/metrics/enso/),        extratropical        modes        of        variability
(https://pcmdi.llnl.gov/metrics/variability_modes/),   monsoon   (https://pcmdi.llnl.gov/metrics/monsoon/),   MJO
(https://pcmdi.llnl.gov/metrics/mjo/), and precipitation benchmarking (https://pcmdi.llnl.gov/metrics/precip/). We
plan to expand this capability to other metrics in the PMP, such as the cloud feedback analysis. The majority of the



PMP's interactive plots have been developed using Bokeh (https://bokeh.org/), a Python data visualization library that
enables the creation of interactive plots and applications for web browsers.
**4 Model Benchmarking**
While the PMP originally focused on evaluating multiple models (e.g., Gleckler et al., 2008), in parallel there has
been increasing interest from model developers and modeling centers to leverage the PMP to track performance
evolution in the model development cycle, as discussed in Gleckler et al. (2016). For example, metrics from the PMP
have been used to document performance of ESMs developed in the U.S. DOE Exascale Earth System Model (E3SM;
Caldwell et al., 2019; Golaz et al., 2019; Rasch et al., 2019; Hannah et al., 2021; Tang et al., 2021), NOAA
Geophysical Fluid Dynamics Laboratory (GFDL; Zhao et al., 2018), Institut Pierre-Simon Laplace (IPSL; Boucher et
al., 2020; Planton et al., 2021), National Institute of Meteorological Sciences-Korea Meteorological Administration
(NIMS-KMA; Sung et al., 2021), University of California, Los Angeles (Lee et al., 2019b), and the Community
Integrated Earth System Model (CIESM) project (Lin et al., 2020).
To make the PMP more accessible and useful for modeling groups, efforts are underway to broaden workflow
options. Currently, a typical application involves computing a particular class of performance metrics (e.g., mean
climate) for all CMIP simulations available via ESGF. To facilitate the ability of modeling groups to routinely use the
PMP during their development process, we are working to provide a customized workflow option to run all the PMP
metrics more seamlessly on a single model, and to compare these results with a database of PMP results obtained from
CMIP simulations (see Code and Data Availability section). Via the PMP-documented and pre-calculated metrics
from simulations in the CMIP archive, it is possible to readily incorporate CMIP results into the assessment of new
simulations, without retrieving all CMIP simulations and recomputing the results. The resulting quick-look feedback
can highlight model improvement (or deterioration) and can assist in determining development priorities or in the
selection of a new model version.
As an example, here, we show PMP results obtained from GFDL-CM3 from CMIP5 and GFDL-CM4 from
CMIP6, for a demonstration of using the Taylor Diagram to compare versions of a given model (Fig. 11). One
advantage of the Taylor Diagram is that it collectively represents three statistics (i.e., centered RMSE, standard
deviation, and correlation) in a single plot (Taylor, 2001), which synthesizes the performance intercomparison of
multiple models (or different versions of a model). In this example, four variables were selected to summarize
performance evolution (shown by arrows) in multiple seasons. Except for boreal winter, both model versions are
nearly identical in terms of net TOA radiation, however in all seasons the longwave cloud radiative effect is clearly
improved in the newer model version. The TOA flux improvements likely contributed to the precipitation
improvements, by improving the balances of radiative cooling and latent heating. The improvement in the newer
model version is consistent with that documented by Held et al., (2019) and evident via the arrow directions pointing
to the observational reference point.
Parallel Coordinate Plots can also be used to summarize the comparison of two simulations for their
performance. In this section, as an example we demonstrate the comparison of selected metrics: the mean climate,
ENSO, and ETMoV (Fig. 12). To facilitate comparison of a subset of models, a few models can be selected and





highlighted as connected lines across individual vertical axes on the plot. With the PMP, a common application is to
select two versions of the same model to contrast their performance (solid lines) against the backdrop of results from
other models, shown as violin plots for the distribution of statistics from other models on each vertical axis. The
spatiotemporal RMSE (i.e., temporally averaged spatial RMSE of annual cycle climatology patterns) is used for mean
climate as discussed in Section 3.1. The PMP's ENSO metrics that were discussed in Section 3.2 and the RMSE
representing total error of ETMoV that were discussed in Section 3.3 are respectively used for ENSO and ETMoV.
The plot is simplified from Figure 2 to more efficiently highlight the difference in performance of two GFDL models:
GFDL-CM3 and GFDL-CM4. Each vertical axis indicates performance for each metric defined for climatology of
variables (Fig. 12a), ENSO characteristics (Fig. 12b), or interannual variability mode obtained from seasonal or
monthly averaged time series (Fig. 12c). In this example, it is shown that GFDL-CM4 is superior to GFDL-CM3 for
most cases across selected metrics (downward arrows in green) while inferior for a few cases (upward arrows in red)
— consistent with previous findings (Held et al., 2019; Planton et al., 2021; Chen et al., 2021). Such applications of
the Parallel Coordinate Plot can enable quick overall assessment and tracking of the ESM performance evolution
during its development cycle. More examples showing other models are available in the Supplementary material (Figs.
S1 to S3).
Note that there have been efforts to coalesce objective model evaluation concepts used in the research
community (e.g., Knutti et al., 2010), however as the field continues to evolve rapidly, definitions are still being
finessed, and there is room for the community to further advance well-established metrics. Via the PMP, we produce
hundreds of summary statistics, but it will not be surprising if only a subset of them might be considered as viable
candidate metrics for more practical routine performance evaluations.

**5 Discussion**
Given the critical role ESMs play in our efforts to understand a changing climate, scientists involved in the analysis
of ESM simulations have been compelled to improve the process of model evaluation. Current progress towards
systematic model evaluation remains dynamic, with evolving approaches and many independent paths being pursued.
This has resulted in the development of diversified model evaluation software packages. For example, ESMValTool
(Eyring et al., 2016, 2019, 2020; Righi et al., 2020) is a comprehensive package led by a European core development
team that has been used for numerous applications including producing model evaluation plots in Chapter 3 of the
IPCC's AR6 Working Group 1 Assessment (Eyring et al., 2021). The Model Diagnostics Task Force (MDTF)
Diagnostics package, led by NOAA, focuses on process-oriented diagnostics (Maloney et al., 2019; Neelin et al.,
2023). The International Land Model Benchmarking (ILAMB) Software System (Collier et al., 2018) led by Oak
Ridge National Laboratory provides land surface and carbon cycle metrics with key state-ot-the art observational
products, and similarly, the International Ocean Model Benchmarking (IOMB) Software System (Fu et al., 2022)
focuses on surface and upper ocean biogeochemical variables. The Climate Variability Diagnostics Package (CVDP;
Phillips et al., 2014; Fasullo et al., 2020) developed at NCAR provides diagnosis of climate modes of variability.
Analyzing Scales of Precipitation (ASoP; Klingaman et al., 2017; Martin et al., 2017; Ordonez et al., 2021) focuses
on analyzing precipitation scales across space and time. In parallel, the regional climate community also has actively



developed metrics packages such as the Regional Climate Model Evaluation System (RCMES; Lee et al., 2018a;
Whitehall et al. 2012). Separately, a few climate modeling centers have developed their own model evaluation
packages to assist in their in-house ESM development, e.g., the E3SM Diags (Zhang et al., 2022). There also have
been other efforts to enhance the usability of in-situ and field campaign observations in ESM evaluations, such as
Atmospheric Radiation Measurement (ARM) GCM Diag (Zhang et al., 2018, 2020) and Earth System Model Aerosol–
Cloud Diagnostics (ESMAC Diags; Tang et al., 2022, 2023).
The model evaluation packages currently being advanced within the ESM research community all have their
own technical approaches and scientific priorities. We believe that this diversity has made, and will continue to make,
the model evaluation process even more comprehensive and successful. The fact that there is some overlap in a few
cases is advantageous because it enables the cross-verification of results, which is particularly useful in the more
complex analyses. Despite the advantages, having no single best or widely accepted approach for the community to
follow, does introduce complexity to the coordination of model evaluation. To facilitate collective usages of individual
evaluation tools, the CMEC has initiated the development of a unified code base that technically coordinates the
operation of distinct but complementary tools (Ordonez et al. 2021). Currently, the PMP, ILAMB, MDTF and ASoP
have become CMEC-compliant by adopting the common interface standards that define how evaluation tools interact
with observational data and climate model output. We expect that CMEC can also help the model evaluation
community to establish standards for archiving the metrics output, much as the community did for the conventions to
describe climate model data (e.g., CMIP application of CF Metadata Conventions [http://cfconventions.org/]; Hassell
et al., 2017; Eaton et al., 2022).
It is worth noting that the comprehensive database of PMP results offers a resource for exploring the range
of structural errors in CMIP class models and their interrelationships. For example, examination of cross-metric
relationships between mean-state and variability biases can shed additional light on the propagation of errors (e.g.,
Kang et al., 2020; Lee et al., 2021b). There continues to be interest in ranking models for specific applications (e.g.,
Ashfaq et al., 2022; Goldenson et al., 2023; Longmate et al., 2023; Papalexiou et al., 2020) or to "move beyond one
model one vote" in multi-model analysis to reduce uncertainties in the spread of multi-model projections (e.g., Knutti,
2010; Knutti et al., 2017; Sanderson et al., 2017; Herger et al., 2018; Hausfather et al., 2022; Merrifield et al., 2023).
While we acknowledge potential interests in using the results of the PMP or equivalent to rank models or identify
performance outliers (e.g., Sanderson and Wehner, 2017), we believe the many challenges associated with model
weighting are application dependent, and thus leave it up to users of the PMP to make those judgments.

**6 Summary and Future Directions**
The PMP has provided quasi-operational ESM evaluation capabilities that can be rapidly deployed to objectively
summarize a diverse suite of model behavior with results made publically available. This can be of value in the
assessment of community intercomparisons like CMIP, the evaluation of large ensembles, or the model development
process. By documenting objective performance summaries produced by the PMP and making them available via
detailed version control, additional research is made possible beyond the baseline model evaluation, model
intercomparison, and benchmarking. The outcomes of PMP's calculations applied to the CMIP archive culminate in



the PCMDI Simulation Summary (https://pcmdi.llnl.gov/metrics/). This summary serves as a comprehensive
repository of PMP outputs, visually capturing the outcomes of objective model-to-observation comparisons. Special
attention is dedicated to the most recent ensemble of models contributing to CMIP6. By offering a comprehensive
assessment of simulated climate, its variability modes, and characteristics of precipitation in ESMs, the PMP
framework equips model developers with quantifiable benchmarks to validate and enhance model performance.
With the growing interest in augmenting the suite of metrics within PMP that reflects an evolving landscape
of evaluation needs, continual efforts are channeled into expanding the scope of the PMP. For example, in coordination
with the World Meteorological Organization (WMO)'s WGNE MJO Task Force, additional candidate MJO metrics
for PMP inclusion have been identified to facilitate more comprehensive assessments of the MJO. Implementation of
metrics for MJO amplitude, periodicity, and structure into the PMP is planned. The ongoing collaboration with NCAR
aims to incorporate metrics related to the upper atmosphere, specifically the Quasi-Biennial Oscillation (QBO) and
QBO-MJO metrics (e.g. Kim et al., 2020). We also have plans to grow the scope of PMP beyond its traditional
atmospheric realm to include domains like the ocean and Arctic regions through collaboration with the U.S. DOE's
project entitled High Latitude Application and Testing of ESMs (HiLAT, https://www.hilat.org/). This dimension of
evaluation holds promise in offering deeper insights into model performance.
In addition to the scientific challenges associated with diversifying objective summaries of model
performance, there are numerous potential areas to advance accompanying technologies, in large part related to the
rapidly evolving set of open-source tools and methods available to scientists. We expect that the current ongoing PMP
code modernization effort to fully adapt the xCDAT will potentially galvanize greater community involvement. We
will continue to maintain robust rigorousness in the calculation of statistics for the PMP metrics by staying tuned with
the latest progress in the field, such as implementing the method for more rigorous conservation in horizontal
interpolation (Taylor, 2023). To improve clarity of key deliverable messages from multivariate data of PMP's metrics
obtained from comprehensive ESM archives, we will consider implementing the advances in the high-dimensional
data visualization field, such as the circular plot discussed in Lee et al. (2018b) and variations of Parallel Coordinate
Plots proposed by Hassan et al. (2019) and Lu et al. (2020).
Looking ahead, the PMP framework is also poised to contribute to high-resolution climate modeling
communities, notably the High Resolution Model Intercomparison Project (HighResMIP; Haarsma et al., 2016) and
the DYnamics of the Atmospheric general circulation Modeled On Non-hydrostatic Domains (DYAMOND; Stevens
et al., 2019). This motivates developments of specialized metrics for high-resolution models, which demonstrate the
features that high-resolution models have enabled. Potential avenue of exploration involves leveraging Machine
Learning (ML) techniques, considering the examined applicability of ML and other state-of-the art data science
techniques being used for process-oriented ESM evaluation works (e.g., Nowack et al., 2020; Labe and Barnes, 2022;
Dalelane et al., 2023). Applications of ML detections, such as for storms using TempestExtremes (Ullrich and
Zarzycki 2017; Ullrich et al., 2021) and fronts (e.g, Biard and Kunkel, 2019), can enable additional specialized storm
metrics for high resolution simulations. For convection permitting models, yet more storm metrics can be applied such
as Mesoscale convective systems. Into the PMP, we currently have plans to implement atmospheric blocking metrics
that were developed through the collaboration of Colorado State University and the PCMDI (Valkonen et al., in prep),



and Atmospheric River detection metrics that are currently under development at LLNL. Both of these metrics suites were developed using the pattern detection capabilities in the latest TempestExtremes (Ullrich et al., 2021). This application of the PMP aligns with a broader plan for regional expansion, with a deliberate emphasis on processes intrinsic to specific regions.

We anticipate that the PMP will continue to play a crucial role in benchmarking ESMs in the future. Improvements in PMP, coupled with advancements in projects within the MIP community, will significantly contribute to assessing the evolving performance of ESMs including via the collaboration with the CMIP Benchmarking Task Team. Enhancements in version control and transparency within obs4MIPs are poised to enhance the provenance and reproducibility of PMP results, thereby strengthening the foundation for rigorous and repeatable performance benchmarking. The PMP's collaboration with the CMIP Forcing Task Team, through the Input4MIPs (Durack et al., 2018) and the CMIP6Plus projects, will further expand the utility of performance metrics in identifying problems associated with the forcing dataset and their application and use in reproducing the observed record of historical climate. Furthermore, as ESMs advance towards more operationalized configurations to meet the demands of decision-making processes (Jakob et al., 2023), the PMP holds significant potential to provide interoperable ESM evaluation and benchmarking capabilities to the community.

**Author Contributions**

All authors contributed to the design and implementation of the research, analysis of the results, and to writing of the manuscript. All authors contributed to the development of codes/metrics in the PMP, its ecosystem tools, and/or the establishment of use cases. JL and PJG led and coordinated the paper with input from all authors.

**Code and Data Availability**

The source code of PMP (Lee et al., 2023b) is available as an open-source Python package: https://github.com/PCMDI/pcmdi_metrics (last access: 21 November 2023) with versions archived on Zenodo DOI: https://doi.org/10.5281/zenodo.592790 (last access: 21 November 2023). The PMP results database (Lee et al., 2023a) that includes calculated metrics is available on the GitHub repository at https://github.com/PCMDI/pcmdi_metrics_results_archive (last access: 21 November 2023) with versions archived on Zenodo DOI: https://doi.org/10.5281/zenodo.10181201. The interactive visualizations of the PMP results are available on the PCMDI website at https://pcmdi.llnl.gov/metrics (last access: 21 November 2023). The CMIP5 and CMIP6 model outputs and obs4MIPs datasets used in this paper are available via the Earth System Grid Federation at https://esgf-node.llnl.gov/ (last access: 21 November 2023).

**Competing interests**

At least one of the (co-)authors is a member of the editorial board of *Geoscientific Model Development*.



**Acknowledgment**

We acknowledge the World Climate Research Programme, which, through its Working Group on Coupled Modeling, coordinated and promoted CMIP6. We thank the climate modeling groups for producing and making available their model output, the Earth System Grid Federation (ESGF) for archiving the data and providing access, and the multiple funding agencies that support CMIP6 and ESGF. This work is performed under the auspices of the U.S. DOE by Lawrence Livermore National Laboratory (LLNL) under Contract No. DE-AC52-07NA27344. Efforts of JL, PJG, MA, AO, PU, KET, PD, CB, MDZ, LC, and BD were supported by the Regional and Global Model Analysis (RGMA) program of the U.S. Department of Energy (DOE) Office of Science (OS), Biological and Environmental Research (BER) program. MFW was supported by the Director, OS, BER of the U.S. DOE through the RGMA program under Contract No. DE340AC02-05CH11231. AGP was supported by U.S. DOE through BER RGMA through Award Number DE-SC0022070 and via National Science Foundation (NSF) IA 1947282, and by National Center for Atmospheric Research (NCAR), which is a major facility sponsored by the NSF under Cooperative Agreement No. 1852977. YYP and EG were supported by the Agence Nationale de la Recherche ARISE project, under Grant ANR-18-CE01-0012, and the Belmont project GOTHAM, under Grant ANR-15-JCLI-0004-01, the European Commission's H2020 Programme "Infrastructure for the European Network for Earth System Modelling Phase 3 (IS-ENES3)" project under Grant Agreement 824084. DK was supported by the New Faculty Startup Fund from Seoul National University and the KMA R&D program (KMI2022-01313). The authors thank Program Manager Renu Joseph of the U.S. DOE for the support and advocacy for the Program for Climate Model Diagnosis and Intercomparison (PCMDI) project and the PMP. We thank Stephen Klein for his leadership for the PCMDI project from 2019 to 2022. We acknowledge contributions from our LLNL colleagues, Lina Muryanto and Zeshawn Shaheen (Now at Google LLC) during the early stage of the PMP, and Sasha Ames, Jeff Painter, Chris Mauzey, and Stephen Po-Chedley for the PCMDI's CMIP database management. The authors also thank Liping Zhang for her comments during GFDL's internal review process.

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





**Table 1.** List of variables and observation datasets used as reference datasets for the PMP's
mean climate evaluation in this paper (Section 3.1 and Figs. 1-2). A ditto mark (") indicates the
same as above.

| Variable | Variable full name | Product | Reference |
|---|---|---|---|
| ps | Precipitation | GPCP-2-3 | Adler et al. (2018) |
| psl | Sea level pressure | ERA-5 | Hersbach et al. (2020) |
| rlds | Surface Downwelling Longwave Radiation | CERES-EBAF-4-1 | Loeb et al. (2018) |
| rltcre | Longwave cloud radiative effect | " | |
| rlus | Surface Upwelling Longwave Radiation | " | |
| rlut | Upwelling longwave at the top of atmosphere | " | |
| rsds | Surface Downwelling Shortwave Radiation | " | |
| rsdt | TOA Incident Shortwave Radiation | " | |
| rstcre | Shortwave cloud radiative effect | " | |
| rsut | Upwelling shortwave at the top of atmosphere | " | |
| rt | Net radiative flux | " | |
| ta-200, ta-850 | Air temperature at 850 and 200 hPa | ERA-5 | Hersbach et al. (2020) |
| tas | 2-m air temperature | " | |
| tauu | Surface zonal wind stress | ERA-INT | Dee et al. (2011) |
| ts | Surface temperature | ERA-5 | Hersbach et al. (2020) |
| ua-200, ua-850 | Zonal wind component at 850 and 200 hPa | " | |
| va-200, va-850 | Meridional wind component at 850 and 200 hPa | " | |
| zg-500 | Geopotential height at 500 hPa | " | |





**Figure 1**. Portrait plot for spatial RMSE (uncentered) of global seasonal climatologies for (a)
CMIP5 (models ACCESS1-0 to NorESM1-ME on the ordinate) and (b) CMIP6 (models
ACCESS-CM2 to UKESM1-1-LL on the ordinate) for 1981-2005 epoch. The RMSE of each
variable is normalized by the median RMSE of all CMIP5 and 6 models. A result of 0.2 (-0.2) is
indicative of an error that is 20% greater (lesser) than the median RMSE across all models.
Models in each group are sorted in alphabetical order. Full names of variable names on the
abscissa and their reference datasets can be found in Table 1. Detailed information for models
can be found at the *Earth System Documentation* (ES-DOC, https://search.es-doc.org/; Pascoe



et al., 2020). The interactive version of the Portrait plot in this figure is available on the PMP
result pages on the PCMDI website (https://pcmdi.llnl.gov/metrics/mean_clim/).



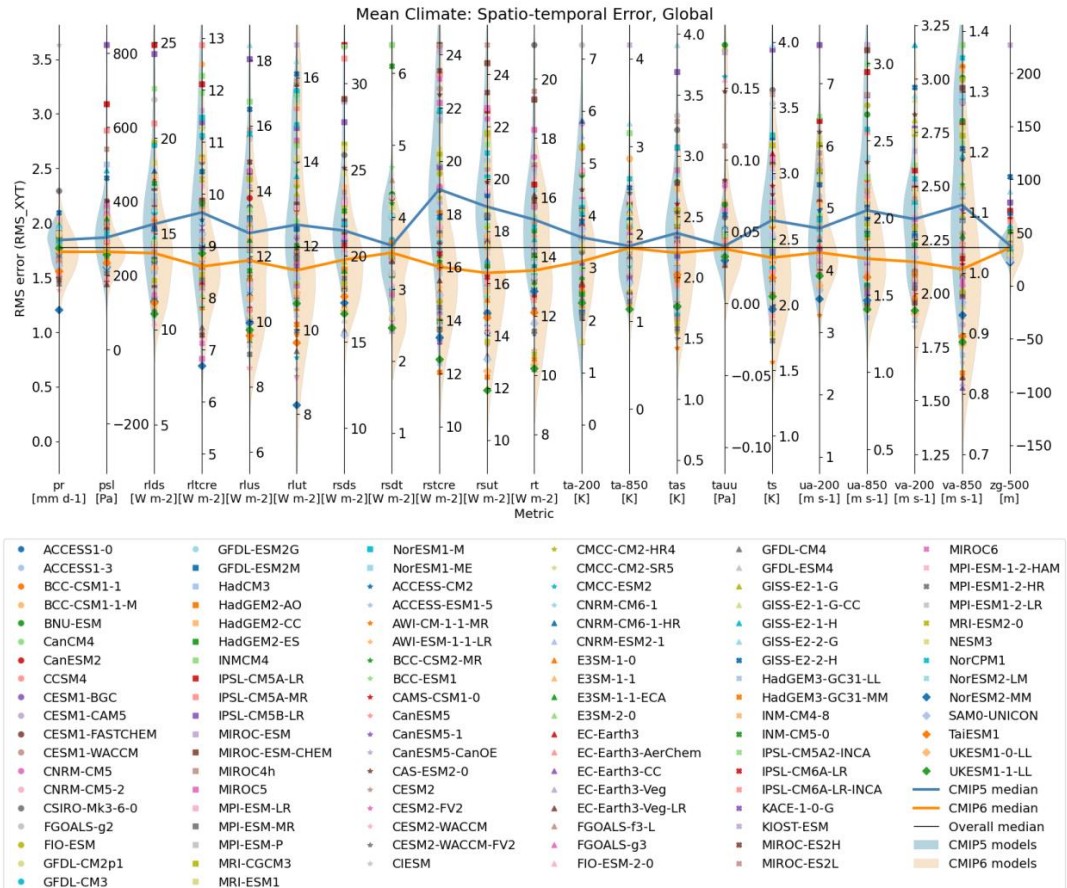

**Figure 2.** Parallel Coordinate Plot for spatio-temporal RMSE (Gleckler et al., 2008) from mean climate evaluation. Each vertical axis represents a different variable. Results from each model are displayed as symbols. Middle of each vertical axis is aligned with the median statistic of all CMIP5 and CMIP6 models. The cross-generation model distributions of model performance are shaded on the left (CMIP5, blue) and right (CMIP6, orange) sides of each axis. Also, medians from CMIP5 (blue) and CMIP6 (orange) model groups are highlighted as lines. Full names for model variables on the abscissa and their reference datasets can be found in Table 1. Time epoch used for this analysis is 1981-2005. Detailed information for models can be found at the *Earth System Documentation* (ES-DOC, https://search.es-doc.org/; Pascoe et al., 2020). The interactive version of the Portrait plot in this figure is available on the PMP result pages on the PCMDI website (https://pcmdi.llnl.gov/metrics/mean_clim/).





**Figure 3.** Application of ENSO metrics to CMIP6 models. Model names with an asterisk (*) indicate that 10 or more ensemble members were used in this analysis. Dots indicate metric values from individual ensemble members while bars indicate the average of metric values across the ensemble members. Bars colored for easier identification of model names at the bottom of the figure. Metrics were grouped into three *Metrics Collections:* (a-n) ENSO Performance, (o-r) ENSO Teleconnections, and (s-w) ENSO processes. Names of individual metrics and default reference datasets being used are noted on top of each panel, and observational uncertainty by applying the metrics for alternative reference datasets noted on the upper right of each panel is shown as gray-shaded. Detailed descriptions for each metric can be found at https://github.com/CLIVAR-PRP/ENSO_metrics/wiki.





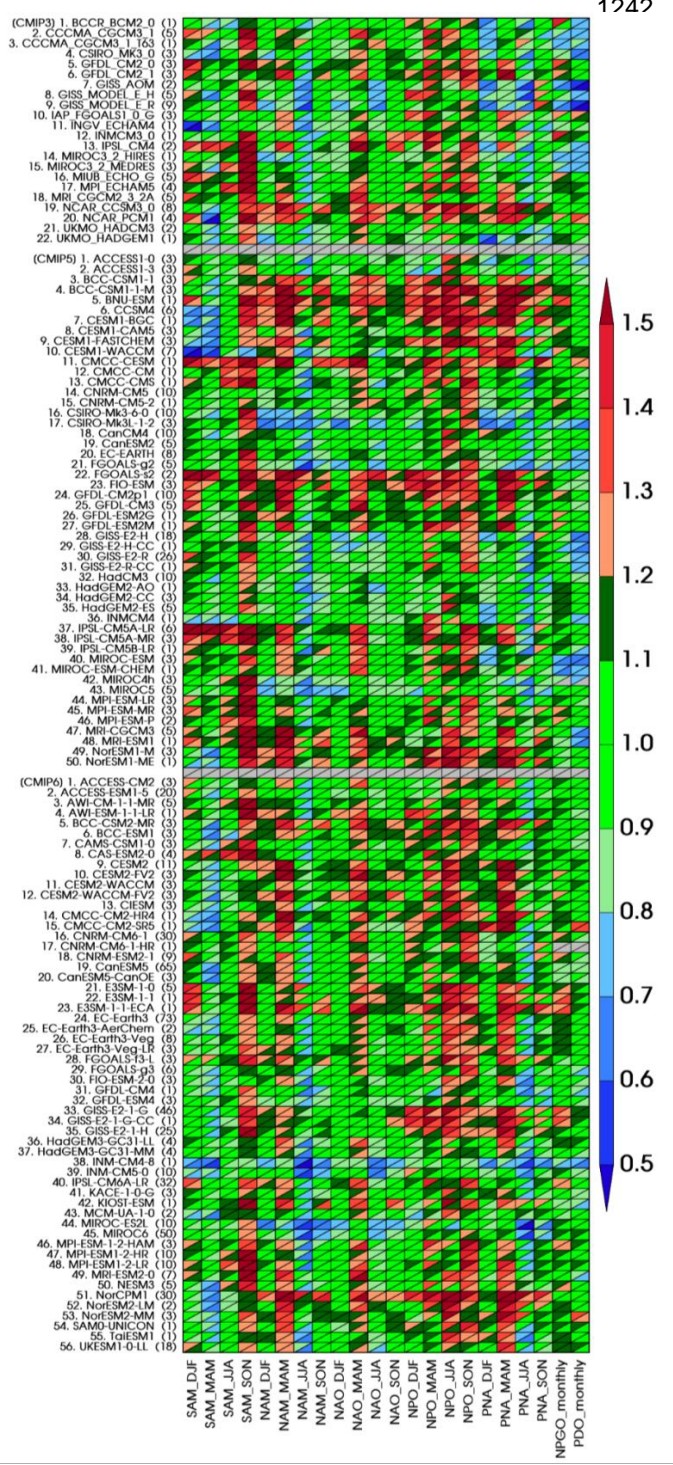

**Figure 4.** Portrait plots of the amplitude of extratropical modes of variability simulated by CMIP3, 5, and 6 models in their historical or equivalent simulations, as gauged by the ratio of spatiotemporal standard deviations of the model and observed PCs, obtained using the CBF method in the PMP. Columns (horizontal axis) are for mode and season, and rows (vertical axis) are for models from CMIP3 (top), CMIP5 (middle), and CMIP6 (bottom), separated by rows of gray boxes. For sea level pressure–based modes (SAM, NAM, NAO, NPO, and PNA) in the upper-left hand triangle the model results are shown relative to NOAA-20CR whereas in the lower-right triangle, the model results are shown relative to the ERA-20C. For SST-based modes (NPGO and PDO), results are shown relative to HadISSTv1.1 (upper-left triangle) and HadISSTv2.1 (lower-right triangle). Numbers in parentheses following model names indicate the number of ensemble members for the model. Metrics for individual ensemble members were averaged for each model. The figure is adapted from Lee et al. 2021b.



(a) Observation

(b) Model

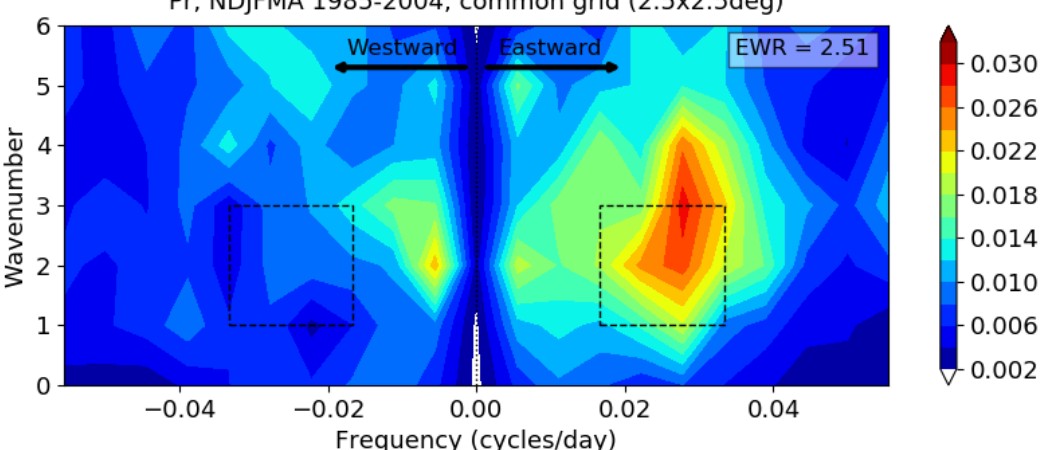

**Figure 5.** MJO EWR diagnostics – wavenumber-frequency power spectra – from (a) GPCP v1.3
(Huffman et al., 2001) and (b) IPSL-CM5B-LR model of CMIP5. The EWR is defined as the ratio
of eastward power (averaged in the box on the right) to westward power (averaged in the box
on the left) from the 2-dimensional wavenumber-frequency power spectra of daily 10°S–10°N
averaged precipitation in November to April (shaded, $mm^2$ $day^{-2}$). Power spectra are calculated
for each year and then averaged over all years of data. The units of power spectra for the
precipitation is $mm^2$ $day^{-2}$ per frequency interval per wavenumber interval.





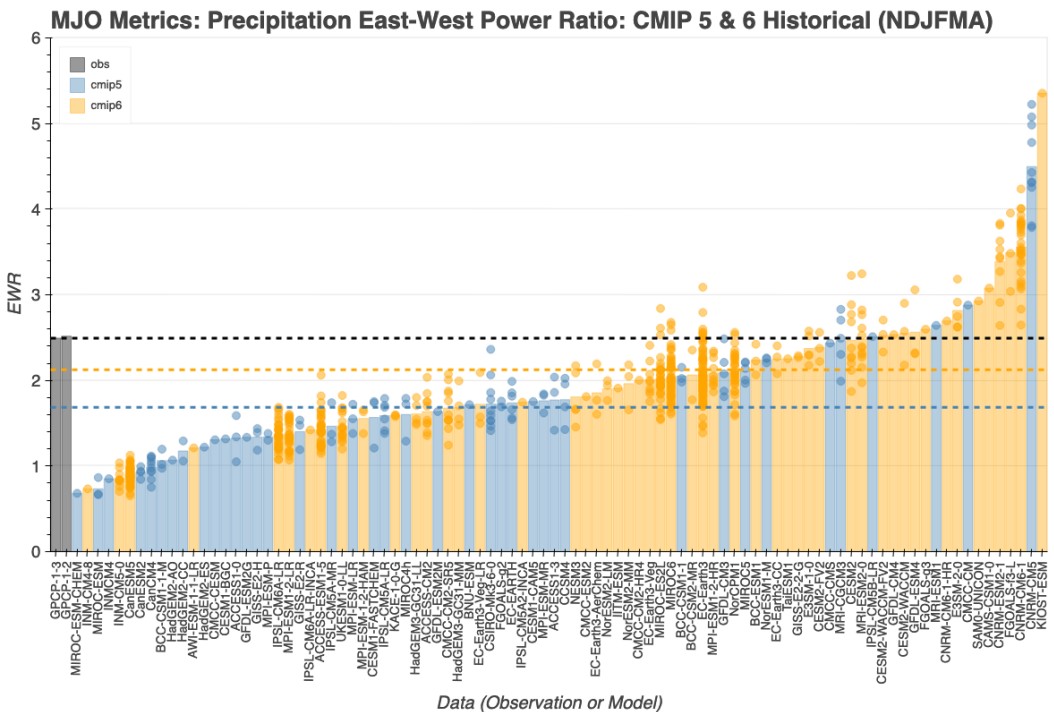

**Figure 6.** MJO EWR from CMIP5 and CMIP6 models, models in two different groups (CMIP5: blue, CMIP6: orange) are sorted by the value of the metric and compared to two observation datasets (purple, GPCP v1.2 and v1.3; Huffman et al., 2001). Horizontal dashed lines indicate EWR from the default primary reference observation (i.e., GPCP v1.3, black), averages of CMIP5 and CMIP6 models. The interactive plot is available at https://pcmdi.llnl.gov/research/metrics/mjo/ where the horizontal axis can be resorted by CMIP group or model names as well. Hover mouse over boxes will show tooltips for metric values and a preview of dive-down plots that are shown in Figure 5.



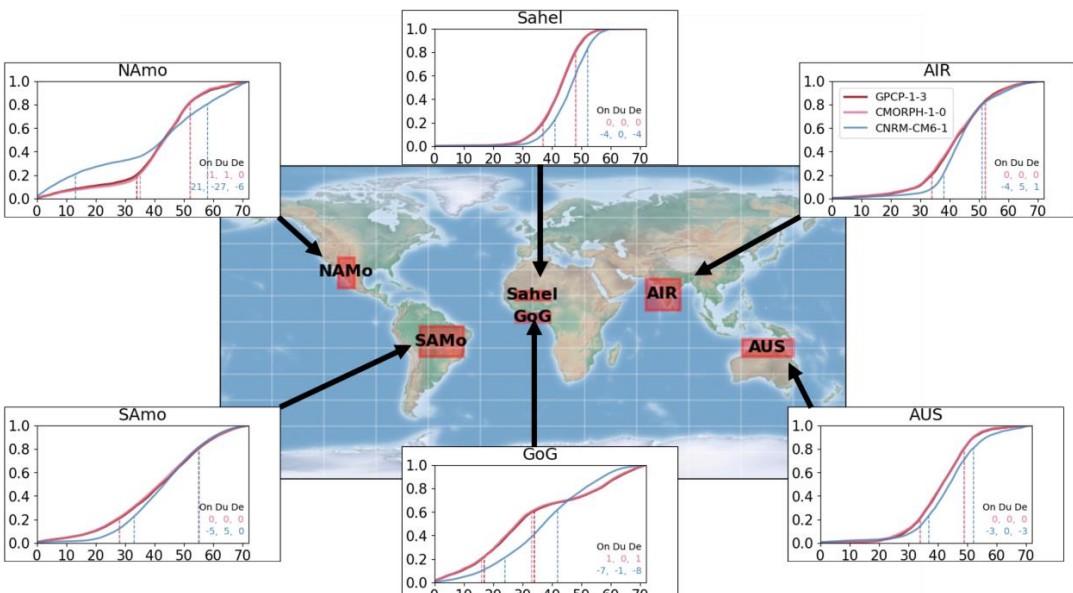

**Figure 7.** Demonstration of the monsoon metrics obtained from observation datasets (GPCP v1.3 and CMORPH v1.0 (Joyce et al., 2004; Xie et al., 2017)) and a CMIP6 model's Historical simulation conducted using CNRM-CM6-1. The results are obtained for monsoon regions: All-India Rainfall (AIR), Sahel, Gulf of Guinea (GoG), North American Monsoon (NAM), South American Monsoon (SAM), and Northern Australia (AUS). The regions are defined in Sperber and Annamalai (2014). Metrics for onset (On), Duration (Du), and Decay (De) derived as differences to the default observation (GPCP v1.3) in pentad indices (observation minus model) are shown at lower right of each panel. Pentad indices for onset and decay of each region are also shown as vertical lines.






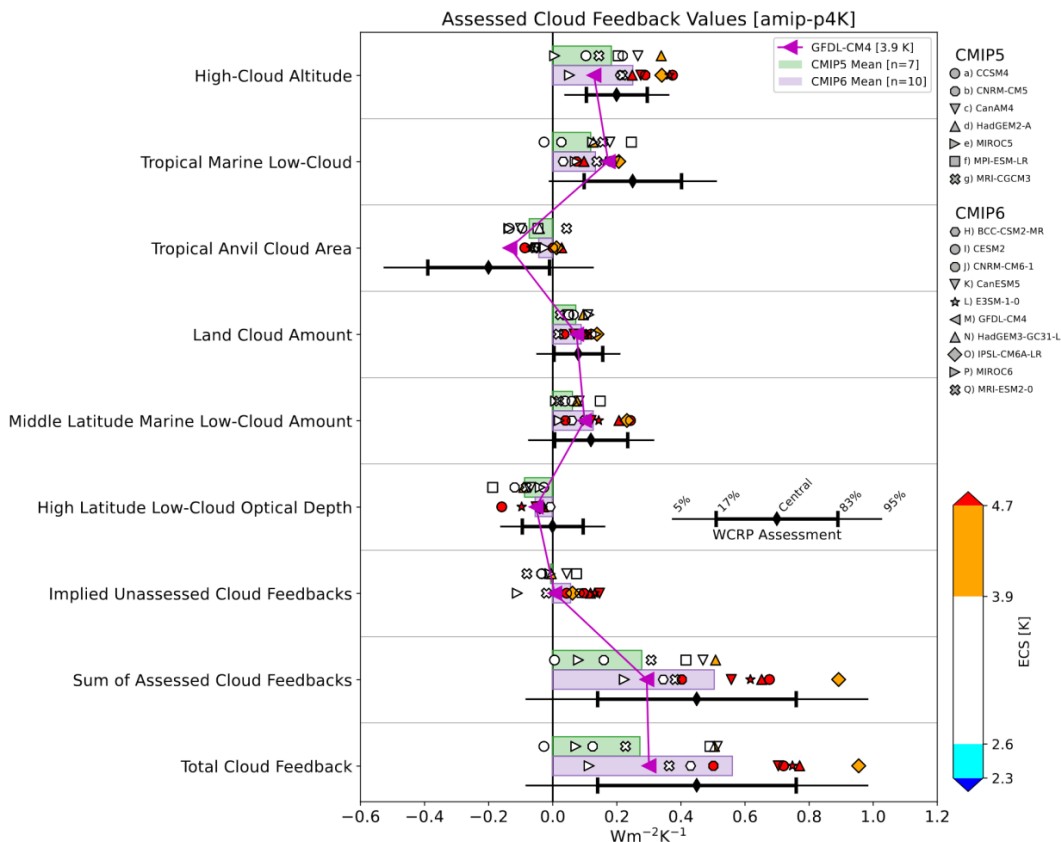


**Figure 8.** Cloud feedback components estimated in amip-p4K simulations from CMIP5 and
CMIP6 models. Symbols indicate individual model values, while horizontal bars indicate multi-
model means. Each model is color-coded by its ECS, with color boundaries corresponding to the
likely and very likely ranges of ECS as determined in Sherwood et al (2020). Each component's
expert-assessed likely and very likely confidence intervals are indicated with black error bars. An
illustrative model (GFDL-CM4) is highlighted.




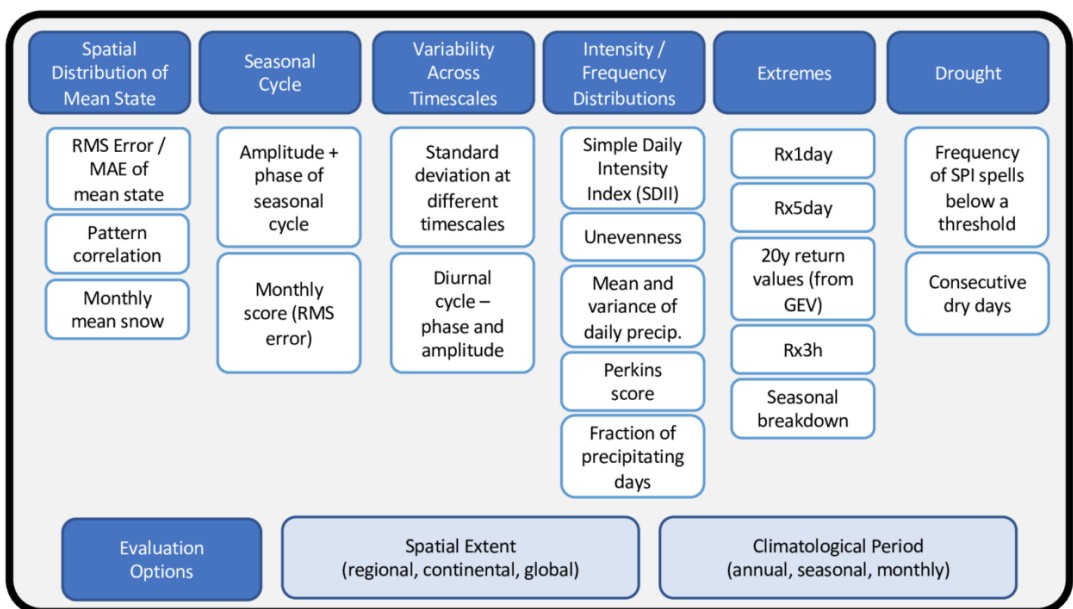

**Figure 9.** Proposed suite of baseline metrics for simulated precipitation benchmarking (figure
reprinted from workshop report; US DOE, 2020).





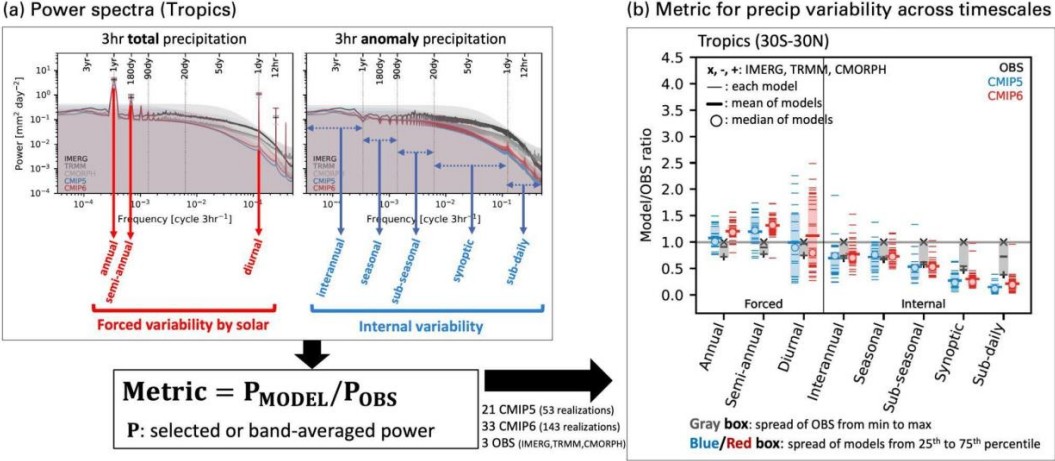

**Figure 10.** Example (a) an underlying diagnostic and (b) its resulting metrics for precipitation variability across timescales. (a) Power spectra of 3-hourly total (left) and anomaly (right) precipitation from IMERG (black), TRMM (gray), CMORPH (silver), CMIP5 (blue), and CMIP6 (red) averaged over the tropics (30°S-30°N). The colored shading indicates the 95% confidence interval for each observational product and for the CMIP5 and CMIP6 means. (b) Metrics for forced and internal precipitation variability based on power spectra. The reference observational product displayed is GPM IMERG (Huffman et al., 2015). The gray boxes represent the spread of the three observational products ("X" for IMERG, "-" for TRMM, and "+" for CMORPH) from the minimum to maximum values. Blue and red boxes indicate the 25th to the 75th percentile of CMIP models as a measure of spread. Individual models are shown as thin dashes, the multimodel mean as a thick dash, and the multimodel median as an open circle. Details for the diagnostics and metrics are described in Ahn et al. (2022).





**Figure 11.** Taylor Diagram contrasting performance of an ESM in their two different versions
(i.e., GFDL-CM3 from CMIP5 and GFDL-CM4 from CMIP6) in its Historical simulation for
multiple variables (precipitation [pr], longwave cloud radiative effect [rltcre], shortwave cloud
radiative effect [rstcre], and total radiation flux [rt]) in their climatology over the globe for (a) DJF,
(b) MAM, (c) JJA and (d) SON seasons. The arrow is directed toward the newer version of the
model from the older version (i.e., GFDL-CM3 → GFDL-CM4).



**Figure 12.** Parallel Coordinate Plot contrasting performance of two different versions of the GFDL model (i.e., GFDL-CM3 from CMIP5 and GFDL-CM4 from CMIP6) in their Historical experiment for errors from (a) mean climate, (b) ENSO, and (c) extratropical modes of variability. Considering lower indicates better, Improvement (degradation) in the later version of the model is highlighted as a downward green (upward red) arrow between lines. Middle of each vertical axis is set to the median of the group of benchmarking models (i.e., CMIP5 and CMIP6), with the axis range stretched to maximum distance to either minimum or maximum from the median for visual consistency. The inter-model distributions of model performance are shown as shaded violin plots along each vertical axis.