# Peer review of "Systematic and Objective Evaluation of Earth System Models: PCMDI"

_EGUsphere, 2023_

## Author Comment (AC1)

**RC1**: 'Comment on egusphere-2023-2720', Anonymous Referee #1, 04 Dec 2023

In their manuscript, Lee et al. present the PCDMI Metrics Package (PMP) version 3, which is an open-source Python software package that provides tools for comparisons of ESMs with each other as well as for comparisons of ESMs with observations. A large range of atmospheric processes can be assessed with this software package. Since this package has been developed within the CMIP comparison projects the PMP results can be produced for all model simulations contributing to CMIP6 and earlier CMIP phases.

The manuscript is generally well written and the package useful for the scientific community, but I have several comments that should be taken into account before publication in GMD. Generally, the authors should take care that their message comes better through. At the moment, I have the feeling that there is a lot written, but what is the take home message? Is this a unique package or are there other, similar packages available. What is new or unique for your specific software package?

We appreciate the reviewer's time for providing thoughtful and constructive comments. Please find our point-by-point response below (text colored in blue). Also, to better emphasize some of the key messages, we have reduced and/or simplified some content throughout the manuscript.

**General comments:**

- This is a quite extensive overview and I was wondering how a new user should get started with PMP. I saw from the links you provided that everything is quite well documented, however, this is so much information at once that I think that a new user will have really trouble getting started. Is there e.g. a documentation that could be downloaded as a single pdf file or does one really spend hours reading everything on the computer screen?

  Thank you for pointing this out. We have an online documentation website (http://pcmdi.github.io/pcmdi_metrics/) that includes instructions for installation and demo for quick start. We have added the link to it in the "Code and Data Availability" section of the revised manuscript. In addition, we also have added a brief description for the installation and also link to the instructions in Section 2 to respond to this comment. We also note that a typical user of the PMP may only make use of a few of the performance metrics available, and thus only need to read the documentation relevant for their purposes.

- How can a user install PMP on a computer? What requirements are needed? You provide a link to your github repository, where this information is provided, but I think this information should also be provided in the manuscript itself.

  As responded above, we have added the installation information in the manuscript, and also clarified the link to the instruction. We agree with the reviewer that it is helpful to point to this explicitly in the manuscript, but we would rather not add too many details.

The reason for this is that the installation requirements periodically need to be updated, and having pointers to the location of the installation documentation (rather than elaborating on technical details) can minimize content in the manuscript becoming obsolete.

- A general questions I was wondering about is, if you are offering workshops for training of new users?

  We appreciate the reviewers' interest. We have provided demo notebooks that users can follow for self-guided tutorials, which are available from the online instruction and we have clarified this in the revised manuscript. In parallel, we are in the process of preparing such training sessions but dates and platforms are yet to be determined. Once decided, we will post it via the PMP website and other available channels to reach out to the community. We plan to provide recorded video of the tutorial via the PMP's online documentation website in the future for those who missed participating.

- In the abstract you state what you will discuss the history up to date, recent updates and future directions. The future directions are discussed to some part in Section 6, but I could not find any information or discussion on the history of PMP up to date or what the updates between former and this version were.

  This information has been described at the beginning of Section 3 in the original submission, but to make this more prominent we have revised the paragraph to further clarify which updates have been made to the latest version of the PMP, and more generally to the origins of the PMP. We also have reorganized the Introduction section to more clearly describe some of the PMP's history.

- Generally, all sections seem to be a bit too lengthy and to my opinion not really coming to the point. Best example is the summary and future directions section. Although you have a discussion section, you provide a quite long summary and future directions section without really summarizing what you have presented.

  We acknowledge the reviewer's point. To respond to this constructive feedback, we have reorganized the discussion and summary sections. In the revised manuscript, we have renamed the last two sessions as "Discussion" and "Summary and Conclusion", and some discussion pertaining to the future directions were moved to the "Discussion" section to make the "Summary and Conclusion" more clear and concise.

- Is this the only tool for analyzing CMIP data or have there also other tools been developed?

  In the original submission we discussed other tools that are available in the community in the first paragraph of the Discussion section. To improve clarity and readability and to respond to the comment, we have moved the description of other fellow tools to the

Introduction section. We also further discussed the diversity of the tools in the following paragraph of the Discussion section:

"Current progress towards systematic model evaluation is exemplified by the diversity of tools being developed (e.g., the PMP, ESMValTool, MDTF, ILAMB, IOMB, and other packages). Each of these tools has its own scientific priorities and technical approaches. We believe that this diversity has made, and will continue to make, the model evaluation process even more comprehensive and successful. The fact that there is some overlap in a few cases is advantageous because it enables the cross-verification of results, which is particularly useful in more complex analyses. Despite possible advantages, having no single best or widely accepted approach for the community to follow, does introduce complexity to the coordination of model evaluation."

**Specific comments:**

P2, L47-48: When was version 1 developed (published)? What are the major changes/new developments you are presenting here?

The first version of the PMP was released in 2015 (https://doi.org/10.5281/zenodo.13673), and we have included this information as a part of the Introduction section in the revised manuscript.

To respond to the reviewer's concern for the lengthy paper from another comment, we have decided to focus more on current and future parts than the history part. Therefore, we have revised the text as follows. "In this paper, we provide an overview of the PMP including its latest capabilities, and discuss its future direction."

P3, L82: Also here you should clearly state when the first version was developed. Has this version been somewhere documented/published? Or has this version just been provided to the CMIP community?

This information has been described in the Code and Data Availability section. To further clarify, we have made the following revisions.

The original text, "To respond to the need, PCMDI has developed the PCMDI Metrics Package (PMP), to quantitatively synthesize results from the archive of CMIP simulations via performance metrics that help characterize the overall agreement between models and observations (Gleckler et al., 2016)" was citing Gleckler et al., 2016 for the first version of the PMP development.

To further clarify, we revised the sentence as follows: "To respond to the need, PCMDI developed the PCMDI Metrics Package (PMP) **and released its first version in 2015 (see Code and Data Availability section for all versions)**. A centralizing goal of the PMP then and now is to quantitatively synthesize results from the archive of CMIP simulations via performance metrics that help characterize the overall agreement between models and observations (Gleckler et al., 2016)."

We also added the following description in the Code and Data Availability section.

"PMP is available as an open-source Python package with **all released** versions archived on Zenodo DOI: https://doi.org/10.5281/zenodo.592790"

P4, L86: Here, some examples should be given. Which performance metrics or statistical measures are used?

To respond to the reviewer's comment, we have added the following description. "Common examples include a domain average bias, a root-mean-square error (RMSE), a spatial pattern correlation, or others, typically selected depending on the application."

P4, L101: Also here, add some examples.

Statistical measures that can be used as metrics are diverse depending on the climate characteristics being evaluated, which we therefore prefer to explain in each subsection of Section 3. Since we have now provided some examples in response to the comment above, we have revised the description as "via well-established statistics **those discussed in Section 3**".

P4, L102: Why only a subset? Specify.

The collection of CMIP experiments includes simulations for pre-industrial and historical periods, future projections with different scenarios, and many others. There are only a few CMIP experiments that are particularly well suited for comparison to recent observations. The most important examples of model evaluation via comparison with observations include the recent historical period as simulated in the CMIP Historical and AMIP experiments. To help clarify the above, we have rewritten the description as follows: "A subset of CMIP experiments, those conducted using the observation forcings such as "Historical" and "AMIP" (Eyring et al., 2016), is particularly well-suited for comparing models with observations." This addition is followed by detailed description for the Historical and AMIP experiments of CMIP.

P4, L102: What do you mean with CMIP class models? Models that participate in CMIP or the kind of models participating in CMIP?

To improve the clarity, we have revised "CMIP-class models" to "CMIP-participating models".

P4, L109: References? Are there any publications?

Yes, and these publications are referenced in each subsection of section 3. However, we also have now also added a few representative publications here to respond to the reviewer's comment.

P6, L170: The following subsections are rather the "processes" that can be assessed with PMP than metrics. The statistics you are using are the metrics for each of this processes.

We understand and appreciate what you have pointed out here, however, after considerable thought we prefer to retain our description. The reason for this is that it is not uncommon to use the word "process" for "process-oriented metrics" in this area of research, and thus to avoid any confusion with it, we would prefer to keep the description as is.

P6, L174: "well-established statistics" -> You should more clearly write here which statistics are used (at least some examples should be given).

We have added the following description: "such as RMSE, mean absolute error (MAE), and pattern correlation"

P6, L178: Provide here a typical example. What is a typical default model and what are the alternatives.

We have added "(e.g., see Table 1)" to clarify the "default" reference datasets.

P6, L189-191: Provide a short explanation how this is done or where this is described.

We have added the following description in the revised manuscript. "Detailed instructions can be found on the PMP's online documentation (http://pcmdi.github.io/pcmdi_metrics)."

P8, L237: Since "Performance", "Processes" and "Teleconnections" are not really metrics, I would suggest to rewrite the sentence as follows: "The ENSO metrics used to assess/evaluate the models are divided into three categories: ………..".

Thank you. Revised accordingly.

P16-18: Summary section is too lengthy and not really summarizing what has been presented in the manuscript. Some part of this should rather be part of the discussion section. Further, there should be a conclusion section e.g. stating clearly what is the gain for the community of this software package. Has it already successful applied for CMIP etc.

To clarify the take home messages, we have reorganized this section. Some discussion of the future directions were moved to the "Discussion" section, and the "Summary" section was renamed to the "Summary and Conclusion" section. We have revised text in both sections to clarify the message.

Figure 1 caption: Add in the figure caption what is shown in the boxes (thus, add that the separation of RMSE by season is shown there)

In the Fig.1 caption, we have added the following description. "The RMSE is calculated for each season (shown as triangles in each box)"

**Technical corrections:**

P4, L108: Add "assessment" or "model" after CMIP.

Revised accordingly.

P8, L247: Abbreviation "ITCZ" has not been introduced.

The full description, "intertropical convergence zone" is added. We also added appendix A that lists the acronyms used in this paper.

P9, L288: Abbreviation "DOE" has not been introduced.

"U.S. Department of Energy (DOE)" is added.

P11, L344: Abbreviation GoG has not been introduced. It is given in the figure caption of figure 7, but not in the text at L338: Should be done there, too.

"GoG" is replaced by "Gulf of Guinea".

P12, L401: Although the abbreviations "WGNE" and "WGCM" have been introduced, I would suggest to repeat it here.

We understand that it would be helpful to re-introduce WGNE and WGCM. However, they have been introduced earlier and the sentence includes multiple other abbreviations, so we decided to not re-introduce those abbreviations. Instead, to address your point we newly added a table of acronyms as an appendix A to help readers.

P12, L 466: Abbreviation "ESGF" has not been introduced.

Earth System Grid Federation (ESGF) was introduced in Section 2 on page 4, but again to improve the readability we have introduced the Table of acronyms.

P15, L493: …….Section 3.3 are respectively -> Section 3.3., respectively, are

Thank you for the suggestion. This sentence was removed in the revised manuscript during the process of condensing, consolidating, and simplifying the text in response to the reviewer's other comment.

P16, L542: use parenthesis instead of brackets

Revised accordingly.

P18, L605: check sentence.

The sentence was rewritten to be more concise and clear.

P41, Figure 6 caption, L1293: units should be given in an upright font.

The metric in Figure 6 is derived as a ratio of powers for eastward and westward wave propagation, thus the metric itself is unitless. To clarify this point, in the caption we indicated that the metric is a ratio and unitless.

---

## Author Comment (AC2)

**RC2**: 'Comment on egusphere-2023-2720', Anonymous Referee #2, 18 Dec 2023

**COMMENTS TO THE AUTHOR(S)**

A comprehensive and objective evaluation of Earth System Models (ESMs) is essential to understand the strengths and weaknesses of individual ESMs and to provide a basis for model improvement. This paper provides an overview of the PCMDI Metric Package (PMP) Version 3, which provides a "quick-look" objective comparison of ESMs to each other and to observations. The purpose, flow, and explanation of the present paper were well written in a concise and easy to understand manner. I expect that the PMP package will be more useful in the context of the CMIP7 project and subsequent phases. I have only a few suggestions before the publication of this paper in the GMD journal.

We appreciate the reviewer's time and their constructive comments. Please find our point-by-point response below, text colored in blue.

**General suggestion:**

I am so curious about how to install the PMP package on my local computer as a user, and what are the requirements for a successful installation. This is the most basic and critical question that readers may have. While the author provides a GitHub link for the related information, I believe it is essential to have a dedicated section in this paper with a brief explanation. This will serve as a catalyst for more readers to become actual PMP users.

We agree that information for installation is important and it is worth including in the paper. To respond, we included the following additional descriptions.

In Section 2:

"The PMP leverages other Python-based open-source tools and libraries. Its installation process is streamlined and user-friendly, leveraging the Anaconda distribution and the conda-forge channel. By employing conda and conda-forge, users benefit from a simplified and efficient installation experience, ensuring seamless integration of PMP's functionality with minimal dependencies. This approach not only facilitates a straightforward deployment of the package but also enhances reproducibility and compatibility across different computing environments, thereby contributing to the accessibility and widespread adoption of PMP within the scientific community. The pointer to the installation instructions can be found in the Code and Data Availability section."

In the Code and Data Availability section:

"PMP's installation process is streamlined using the Anaconda distribution and the conda-forge channel (https://anaconda.org/conda-forge/pcmdi_metrics, last access: 21 February 2024), and the installation instructions are available at http://pcmdi.github.io/pcmdi_metrics/install.html (last access: 21 February 2024)."

We however decided to not include further detailed technical steps in the manuscript because they will likely evolve with technological advances, and we would rather this description not become obsolete. Details are however provided online where they can be updated as needed.

**Minor comments:**

Lines 101-102: I am just wondering if the PMP can be used to evaluate the regional climate models that participated in the RCM project, such as CORDEX. The authors could briefly discuss this or any related future plans (if the authors have any) in the discussion section.

Thank you for this important point. Although the PMP has been traditionally focused on the global scale evaluations, the PMP team has been interested in broadening the PMP's scope to enable regional climate applications. This could be considered in connection with the evaluation of high-resolution models. In the original submission we have discussed this as "*This application of the PMP aligns with a broader plan for regional expansion, with a deliberate emphasis on processes intrinsic to specific regions*" in the "Summary and Future Directions" section of the original manuscript.

In the revised manuscript, we expanded the original text as follows and moved the text to the "Discussion" section to make it more noticeable.

"These example enhancements of the PMP are indicative of an increasing priority to target regional simulation characteristics. With a deliberate emphasis on processes intrinsic to specific regions, this may lead to enabling potential applications of the PMP within the regional climate modeling activities such as Coordinated Regional Downscaling Experiment (CORDEX; Gutowski Jr. et al., 2016)."

Figure 1. Are all model grids (or just land?) used in the evaluation? It would be helpful to indicate this in the figure caption.

Following the comment, we have added the following description to the caption: "The RMSE is calculated over the globe including both land and ocean, and model and reference data were interpolated to a common 2.5x2.5 degree grid."

**For Abbreviation:**

Thank you for pointing out the below missing full names. In addition to responding to each of the comments, we also have added a table of acronyms as Appendix A.

Line 37 (Abstract): Provide the PCMDI's full name.

Revised accordingly.

Line 247. Provide the ITCZ's full name.

The full description, "intertropical convergence zone" is added.

Line 288. Provide the DOE's full name.

"U.S. Department of Energy (DOE)" is added.

Line 344. Provide the GoG's full name.

"GoG" is replaced by "Gulf of Guinea".

Line 466. Provide the ESGF's full name.

Earth System Grid Federation (ESGF) was introduced in Section 2 so we didn't re-introduced it here, but we have added Appendix A for list of acronyms to help improve readability.

---

## Author Response (AR2)

The revised version of the manuscript has significantly improved. I have only some technical corrections that should be considered before publication.

We appreciate the reviewer's time and suggestions for technical corrections. Please find our response below, colored in blue.

P2, L42: "including extremes". What exactly do you mean with extremes? What kind of extremes? Of precipitation? Give an example.

The original sentence, "high-frequency characteristics of simulated precipitation, including extremes", is intended to read as the "extremes" pointing to that of "simulated precipitation." To clarify, we revised the sentence as "its extremes".

P2, L42: The PMP results -> The PMP comparison results

Revised accordingly.

P2, L42: "in the context of all model simulations contributing to…." Not clear, please rephrase. Instead of writing "in the context" I would rather write "using".

Thank you for the suggestion. Revised accordingly.

P2, L43: "priority"? I would rather say "objective" is here the correct term.

Thank you for the suggestion. Revised accordingly.

P2, L45: process? Rather use singular than plural?

Thank you for the suggestion. Revised accordingly.

P3, L53: "……has been an exponential growth of data size…." Mentioning this here makes no sense. You further mention this some lines below ones again, where it makes more sense. Thus, I would suggest to remove this text part here.

Revised accordingly.

P4, L118 and 119: Section should be abbreviated as Sect. except at the begin of the Sentence (see manuscript preparation guidelines). This should also be corrected throughout the manuscript.

Thank you for pointing this out. Revised accordingly throughout the manuscript.

P5, L131: "…protocol constrains the simulation with…." Is here not "prescribes" the correct term rather than "constrains"?

While we consider the terms are interchangeable, we prefer to keep the original sentence, "The AMIP experiment protocol **constrains** the simulation with **prescribed** sea surface temperature (SST)," to avoid repeating of the word "prescribe" in one sentence.

P5, L143: priority -> objective
We revised it to "with this objective in priority"

P6, L188 and 189: "We also release ...." and "are also available online" -> this is a repetition and the sentence should be rephrased.
Thank you for pointing this out. Revised accordingly by removing the "We also release" part.

P6, L189 - 190: Add comma after "statistics" and add a comma before include, so that it reads: "The archive of these statistics, stored as JSON files (.........), includes........".
Thank you for the suggestion. Revised accordingly.

P6, L191: See -> see
Revised accordingly.

P7, L228: delete "at" before regional scales (?)
Thank you for the suggestion. However, we respectfully disagree and prefer to keep it as is because it is indicating "the application of user-custom domains at regional scales."

P7, L230: these -> the (?)
Revised accordingly.

P10, L323: Write SAM_SON without a dash. Even better it would be if you would skip the abbreviation SON since it has not been introduced and is not in the list of abbreviation.
Revised as "SAM in September-October-November (SON)"

P11, L375ff: pentads? I do not see any pentads here, but only squares or rectangles.
I think there might be some confusions. The word "pentad" indicates a "group or set of five", not a "pentagon". Here we use the term to indicate precipitation in 5-day length time chunks obtained from the daily time series, following the metric definition in the reference paper, Sperber and Annamalai (2014). To avoid such confusion, we revised the description as "climatological pentad data" from "climatological pentads".

P13, L435: Figure 8 -> Fig. 8
Revised accordingly.

P13, L450 and P14, L481: Tropics -> tropics
Revised accordingly.

P14, L462: evaluations -> evaluation
Revised accordingly.

P29, L887: reference of Hintze -> remove indent for the first line of the reference.
Revised accordingly.

P30, L893: hydrometeorology -> Hydrometeorology
Revised accordingly.

References in general: Journal names are usually abbreviated (see Copernicus manuscript
preparation guidelines)
Thank you for pointing this out. Revised accordingly.